# Semi-Supervised Joint Learning for Hand Gesture Recognition from a Single Color Image

**DOI:** 10.3390/s21031007

**Published:** 2021-02-02

**Authors:** Chi Xu, Yunkai Jiang, Jun Zhou, Yi Liu

**Affiliations:** 1School of Automation, China University of Geosciences, Wuhan 430074, China; xuchi@cug.edu.cn (C.X.); jchow@cug.edu.cn (J.Z.); 2Hubei Key Laboratory of Advanced Control and Intelligent Automation for Complex Systems, Wuhan 430074, China; 3Engineering Research Center of Intelligent Technology for Geo-Exploration, Ministry of Education, Wuhan 430074, China; 4CRRC Zhuzhou Electric Locomotive Co., Ltd. 1 TianXin Road, Zhuzhou 412000, China; liuyi_hust@163.com; 5National Innovation Center of Advanced Rail Transit Equipment, Zhuzhou 412000, China

**Keywords:** hand gesture recognition, hand pose estimation, joint learning, shared feature

## Abstract

Hand gesture recognition and hand pose estimation are two closely correlated tasks. In this paper, we propose a deep-learning based approach which jointly learns an intermediate level shared feature for these two tasks, so that the hand gesture recognition task can be benefited from the hand pose estimation task. In the training process, a semi-supervised training scheme is designed to solve the problem of lacking proper annotation. Our approach detects the foreground hand, recognizes the hand gesture, and estimates the corresponding 3D hand pose simultaneously. To evaluate the hand gesture recognition performance of the state-of-the-arts, we propose a challenging hand gesture recognition dataset collected in unconstrained environments. Experimental results show that, the gesture recognition accuracy of ours is significantly boosted by leveraging the knowledge learned from the hand pose estimation task.

## 1. Introduction

People interact with each other using hand gestures in everyday life. Hand gesture recognition is an important research topic which has a wide range of applications, such as robotics, human-computer interaction, assistant driving, and so on. Gestures can be classified into two categories: static gesture [1,2,3,4] (which is commonly referred to as “gesture” for short, and in sign languages it is also referred to as “handshape”) and action [5,6,7] (i.e., “dynamic gesture”, and in some papers it is also referred to as “gesture” for short). Different from action recognition which requires video devices or continuous image sequences as input, static gesture recognition requires only a single image as input, thus it can be conveniently and flexibly applied in many scenarios. Furthermore, static gestures are basic components of actions, and static gesture recognition can serve as a key component embedded in action recognition applications. In this work, we focus on the static hand gesture recognition [1,2,3,4]. And the word “gesture” denotes static gesture by default for convenient in the following text.

Hand gesture recognition [1,4] and hand pose estimation [6,8,9] are two closely correlated tasks. A specific hand gesture is commonly associated with a specific hand pose. The hand gesture recognition task focuses on classifying an input image to a gesture category, while the hand pose estimation task explicitly recovers more information, such as positions of finger joints, view point, rotation, scale, and so on. As the hand gesture recognition performance can be affected by the factors related to hand pose, the hand pose information recovered from the input image will be helpful to improve the hand gesture recognition task. However, most of the methods address these two tasks separately, and thus the relationship between the hand gesture and the hand pose are not fully explored. Some methods [2,10,11] recognize the hand gesture directly based on the result of hand pose estimation, but inaccurate hand pose leads to false gesture classification, and the hand gesture recognition performance will be bounded by the upper-limit of the hand pose estimation accuracy.

In this paper, we propose a deep-learning based approach which effectively transfers the hand pose estimation knowledge to the hand gesture recognition task by joint learning an intermediate level shared feature. The shared feature contains not only the information for classifying hand gesture, but also the extra information for predicting hand pose (i.e., relative hand pose, rotation, translation, and scale), which helps improve the gesture recognition accuracy. In our approach, the hand gesture recognition task is not directly based on the hand pose estimation result, but is based on the intermediate level shared feature which contains the information of both the two tasks. The hand gesture can be correctly classified even from images with inaccurate hand pose estimation.

To jointly train a shared feature for both the gesture recognition and the pose estimation tasks is not easy, and the difficulty primarily lies in lack of proper annotation. In the standard joint learning process, both the hand pose and hand gesture annotations are required. However, existing datasets focus on either hand gesture recognition or hand pose estimation, and it is difficult to find a dataset which contains both these two types of annotations.

To tackle this problem, a semi-supervised training scheme is designed to extract the shared feature from hand images with only hand gesture annotation or hand pose annotation. In this manner, the hand pose estimation knowledge learned from the hand pose estimation dataset can be transferred to the hand gesture recognition task. Furthermore, an image reconstruction task is introduced to further benefit the semi-supervised training process. The image reconstruction task encodes the input image to a low dimensional latent code and then reconstruct the image from the code. With this task, the training process can be further benefited from hand images with even no annotation.

Most of the methods recognize static hand gestures in simple constrained environments, (e.g., indoor, simple background, single person or single hand per image, etc.). However, the real life environments are complex and unconstrained. As can be seen in Figure 1, in an unconstrained scene, there may exist many disturbing factors, such as cluttered environments, unrelated people, background hands, and so on. In order to evaluate the hand gesture recognition performance of the state-of-the-arts in real life, we propose a challenging hand gesture recognition dataset in which the images are collected in cluttered environments, and the number of hands per image is up to eight. The dataset contains both foreground hands (which are performing specific gestures) and background hands.

It is noted that some works [7,12] also jointly learn the relationship between gesture and pose, but these works are either focus on actions (dynamic gestures) [7], or human body activity [12]. In [7], the actions (dynamic gestures) is recognized by utilizing temporal hand pose feature which requires video clips (or continuous image sequences) as input. Whereas our approach focuses on static hand gesture recognition, and the shared feature is extracted from a single color image.

In summary, the main contributions of our work are three-fold:We propose a hand gesture recognition approach by joint learning a shared feature for gesture recognition and pose estimation. The proposed approach effectively transfers the hand pose estimation knowledge to the hand gesture recognition task.We design a semi-supervised training scheme to jointly learn the shared feature from hand related datasets. The hand gesture recognition task can be benefited from hand images without hand gesture label, or even without any label.We propose a challenging hand gesture recognition dataset collected in complex unconstrained environments for evaluation purpose. Experimental results show that, the proposed approach outperforms that of the compared methods by a large margin.

The rest of the paper is organized as follows—the related works about hand gesture recognition, hand pose estimation, and hand detection are reviewed in Section 2. Details of the proposed method are illustrated in Section 3. The proposed CUG-Hand dataset is introduced in Section 4, and then the experimental results on CUG-Hand and LaRED datasets are presented in Section 5. The dataset and the related code will be released on https://github.com/waterai12/CUG-Hand-Gesture.

## 2. Related Work

Hand gestures can be recognized from different data sources, such as images [1,4], video clips [5], wearable sensors [13,14], and so on. In this paper, we focus on hand gesture recognition from a single color image, as color image can be conveniently accessed and managed with very low cost. Traditional hand gesture recognition methods primarily utilize hand crafted low-level features, such as SIFT [15,16], image moments [17], Gabor filters [18], and so on. In recent years, deep-learning based methods have significantly boosted the gesture recognition performance. In [17], the gesture is recognized by combining traditional low-level feature and Convolutional Neural Network (CNN) high-level feature. In [19], deep features are extracted from point clouds for SVM classification. In [20] stacked denoising auto-encoders are used to classify gesture category of hand. In [3], the hand gestures are detected and classified by a soft attention mechanism. In [4], a deep CNN-based end-to-end system is proposed to detect hand and recognize the hand gesture. Different from previous works, we train a deep shared feature by exploring the relationship between hand gesture recognition and hand pose estimation.

Hand pose estimation is a task closely related to hand gesture recognition. In the last few years, hand pose estimation from single images [2,6,8,9,21,22] has become a research hotspot. In [23], the hand pose is estimated from monocular color image by 3D hand model fitting. In [9], the 3D hand prior is implicitly learned from a deep CNN network. In [24], an image-to-image translation model is used to generate realistic hand pose data. In [25,26], the hand pose is estimated by exploring the latent space learned by generative model. In [21], the Graph Convolutional Neural Network is used to reconstruct the 3D hand pose and shape. In [22], the hand pose is estimated by structured region ensemble network. However, most previous works study the hand gesture recognition and the hand pose estimation tasks separately, and the relationship between the two is not fully investigated.

To explore the relationship between gesture and pose, many researchers conduct the human body gesture/action recognition directly based on the body pose estimation result [27,28,29,30], as the human body pose can be reliably estimated from input images [28]. Some research works also recognize the hand gesture/action based on the hand pose estimation results. For example, Lie group manifold theory [6], SPD manifold learning [31], Random Forest [2], and LSTM network [10,32] are used to recognize the hand gesture/action based on the hand pose estimation results. However, the bottleneck of these works lies in the performance of the hand pose estimation. Comparing to the body, the hand is much smaller and be with more complex articulations. The pose estimation of the hand is not as reliable as that of the body, and existing works [33,34,35,36,37,38] are normally applied in near range scenario. In practice, accurate hand pose produces accurate gesture classification, and inaccurate hand pose leads to false gesture classification. If the hand gesture is recognized directly based on the hand pose estimation result, the gesture recognition accuracy will be bounded by the upper-limit of the pose estimation accuracy.

Instead of directly using the pose estimation result for recognition, some researchers jointly train the action recognition and the pose estimation tasks. In [39], the action recognition and body pose estimation are learned jointly in a multitask framework. In [40], a hierarchical structure model is used to combine action recognition and pose estimation tasks. In [7], hand action recognition and hand pose estimation are collaboratively learned by exploring the temporal pose feature with multi-order. Different from previous works which focus on action (dynamic gesture) recognition from continuous image sequences, our approach focuses on hand gesture recognition from a single image where the temporal information is unavailable. Besides, the previous works primarily utilize datasets with both action and pose annotations [41], but there exists no dataset containing both static hand gesture and hand pose annotations. To overcome the problem of lacking proper annotation, a semi-supervised joint learning scheme is proposed to effectively learn a shared feature for these two tasks.

Existing hand gesture recognition methods are normally evaluated in constrained environments [42,43] (e.g., simple background, single hand per image, or cropped hand image patches). Some research works [3,4] increase the complexity of the scene in some extent, but still there is only single person/hand per image. In real-life scenario, there may exist many unrelated people and many background hands which make the foreground hand detection difficult. Therefore, it is necessary to detect hand for gesture recognition. In [44], skin color is used for hand detection. In [45], hand is detected by a Support Vector Machine (SVM) classifier based on HOG feature. In [46], the deep and shallow layers are combined for hand detection. In [47,48], hand detection and hand orientation prediction is learned jointly. In [49], hand appearance reconstruction is employed to make the detection model more accurate. In this work, we not only detect hands, but also distinguish the foreground hand which is performing a gesture from the background hands. Furthermore, we proposed a challenging hand gesture recognition dataset captured in unconstrained environment, and the dataset can be used to evaluate the performance of the state-of-the-art.

## 3. Methods

The proposed approach aims at hand gesture recognition from single color images in complex unconstrained environment. It takes a single color image as input, detects the foreground hand, recognizes the gesture of the foreground hand, and estimates the 3D hand pose simultaneously. The output of the approach is the 2D location (bounding box), gesture category, and 3D pose of the detected foreground hand. As can be seen in Figure 2, the proposed approach contains 5 modules: (1) foreground hand detection, (2) shared feature extraction, (3) hand gesture recognition, (4) hand pose estimation, and (5) hand image reconstruction. The network can be trained by a semi-supervised learning scheme when the hand pose annotation or the hand gesture annotation is unavailable. Details of the approach will be addressed as follows.

### 3.1. Foreground Hand Detection

Taking a single color image as input, we detect all hand instances and distinguish the foreground hand instances from the background hand instances. We use FPN [50] as backbone for foreground hand detection. FPN takes the activations of the last 4 stages of ResNet as input, and generates the multi-level feature maps. And then, the Region Proposal Network (RPN) takes the multi-level feature maps as input, and generates a set of region proposals. On each pixel of the feature maps, *K* region proposals are parameterized relative to *K* reference anchors. Following [51], we use three scales and three aspect ratios, yielding K=9, and we adopt parameterizations of region proposal as follows:(1)$$\left\{\begin{array}{c} t_{x}=\frac{x-x_{a}}{w_{a}},t_{y}=\frac{y-y_{a}}{h_{a}},t_{w}=log\frac{w}{w_{a}},t_{h}=log\frac{h}{h_{a}}\\ t_{x}^{*}=\frac{x^{*}-x_{a}}{w_{a}},t_{y}^{*}=\frac{y^{*}-y_{a}}{h_{a}},t_{w}^{*}=log\frac{w^{*}}{w_{a}},t_{h}^{*}=log\frac{h^{*}}{h_{a}}, \end{array}\right.,$$
where x,y denote the two coordinates of the box center, and *w* and *h* denote width and height of the box. Variables $x^{*},x_{a}$ and *x* are for the region proposal box, anchor box, and ground truth box respectively (likewise for y,w,h). A ROI pooling layer extracts feature for each region proposal. And then, the foreground hand prediction step predicts whether the proposals are foreground or background hands, and it further refines the region proposals. After foreground hand prediction, another ROI pooling layer crops the hand image corresponding to the region proposal from the original color image.

The objective function of foreground hand detection is defined as follows:(2)$$L_{detection}=L_{rpn}+L_{fhp},$$
where $L_{rpn}$ is the loss of RPN, and $L_{fhp}$ is the loss of foreground hand prediction.
(3)$$L_{rpn}=\frac{1}{N_{cls}}\sum_{i}L_{cls}\left(p_{i},p_{i}^{*}\right)+\frac{1}{N_{reg}}\sum_{i}p_{i}L_{reg}\left(t_{i},t_{i}^{*}\right).$$

Here, *i* is the index of an anchor, $p_{i}$ is the ground-truth label of whether anchor *i* is a hand, and $p_{i}^{*}$ is the predicted probability. $t_{i}$ is the ground-truth vector representing the 4 parameterized coordinates defined in Equation (Equation 1), and $t_{i}^{*}$ is the corresponding prediction. The classification loss $L_{cls}$ is log loss over two classes (hand vs. non-hand). The regression loss $L_{reg}$ is smooth $L_{1}$ function defined in [52]. The term $p_{i}L_{reg}$ means that the regression loss is activated only for positive anchors $\left(p_{i}=1\right)$ and is disabled otherwise $\left(p_{i}=0\right)$. The loss $L_{fhp}$ is defined similar to $L_{rpn}$, and their difference is that the $L_{cls}$ of $L_{rpn}$ considers two-class classification (hand/non-hand), while the $L_{cls}$ of $L_{fhd}$ considers three-class classification (foreground-hand/background-hand/non-hand).

### 3.2. Shared Feature Extraction

We use a lightweight CNN network [53] as a backbone for shared feature extraction. The network is efficient and accurate, and it is suitable to be adopted into embedded systems such as mobile phone. Following [53], the shared feature extraction network is constructed using basic units named inverted residual block. The intermediate expansion layer in the block uses lightweight depth-wise convolutions.

The foreground hand image patch is resized to a uniform size, and then it is fed into the network for shared feature extraction. The shape of input hand image patch is 256×256×3, in which 256×256 denotes the image size and 3 denotes the number of input channels (color image has 3 channels). The data passes through a series of inverted residual blocks, and the output is of shape 8×8×1280, in which 1280 denotes the number of output channels. An average pooling layer is used to map the output of the network to a shared feature of dimensional 1280. Similar to VAE [54], we generate a latent code with Gaussian distribution. The shared feature is fed to a conv1×1 layer to estimate the parameters of the Gaussian distribution of the latent code, that is, the mean μ and the logarithmic standard deviation σ. And then, a sample *g* is calculated using μ, σ and a standard Gaussian distributed noise Φ as follows:(4)$$g=\mu +\frac{e^{\sigma }}{2}\times \Phi .$$

The mean μ will be used for hand gesture recognition and hand pose estimation, and μ, σ, *g* will be used for image reconstruction. These tasks will be explained in the following three subsections.

### 3.3. Hand Gesture Recognition

We adopt a series of fully connected layers to classify the hand gesture category by using the shared feature. The 1280 dimensional shared feature is converted to the 128 dimensional latent code through a conv1×1. And then the mean μ of the latent code is converted to a 512 dimensional hidden code through a fully connected layer. After the ReLU activation, the 512 dimensional hidden code is converted to a *C* dimensional vector $X={\left[x_{1},x_{2},...,x_{c}\right]}^{T}\in R^{1\times C}$, where *C* denotes the number of categories in the gesture recognition dataset. And then, softmax function is applied on this vector to calculate the score of each gesture category. The gesture with the maximum score is taken as the classification result.

The loss function of hand gesture recognition is defined as the cross entropy between the predicted gesture and the ground truth:(5)$$L_{gesture}\left(X,class\right)=-\log\left(\frac{e^{X_{class}}}{\sum_{i=1}^{C}e^{X_{i}}}\right)=-X_{class}+\log\left(\sum_{i=1}^{C}e^{X_{i}}\right),$$
where class∈(1,2,...,C) represents the gesture category index, and $X_{class}$ represents the score of the predicted gesture with the category index of class.

### 3.4. Hand Pose Estimation

The hand pose is defined by the set of 3D joint coordinates $\left\{P_{i}=\left(x_{i},y_{i},z_{i}\right)\right\}$ with i=1⋯J, in which J=21 denotes the number of hand joints. As the hand size is unknown, to estimate the absolute 3D hand pose from single color image is an ill-posed problem. Following the previous work [9], we estimate the relative 3D hand pose $\left\{P_{i}^{rel}=\left(x_{i}^{rel},y_{i}^{rel},z_{i}^{rel}\right)\right\}$. The length of the first bone of the middle finger of $\left\{P_{i}^{rel}\right\}$ is normalized to a uniform size of 1, and the origin of hand is defined as the root of the middle finger. Let $p_{i}=\left(u_{i},v_{i}\right)$ denote the 2D projection of $P_{i}^{rel}$ in image patch. The projection of the 3D hand joint to 2D image is defined as
(6)$$p_{i}=\Pi \left(R\cdot P_{i}^{rel}\right)\cdot s+t,$$
where $\Pi \left(\cdot \right)$ denotes the 2D projection function. We adopt the orthogonal projection function in this study. R∈SO(3) denotes the 3D rotation of the hand. s∈R and $t\in R^{2}$ denote the scale and the 2D in-plane translation respectively. We define the view parameter V=(R,s,t). The rotation R is parameterized as Euler angles of 3D, and therefore the view parameter V is of 6D.

The mean μ of the latent code generated by the shared feature is fed into the hand pose estimation network to estimate the relative 3D hand pose and the view parameter. The hand pose estimation network contains two fully connected layers, and the activation function is ReLU. The number of neurons in the first hidden layer is 256, and that of the second hidden layer is 128. Two linear layers convert the output of the second hidden layer to the relative 3D hand pose and the view parameter respectively. The loss function of hand pose estimation is defined as follows:(7)$$L_{pose}=L_{rel}+L_{view}=\sum_{i=1}^{J}\|\hat{P_{i}^{rel}}-P_{i}^{rel}{\|}_{2}^{2}+{\left\|\hat{V}-V\right\|}_{2}^{2},$$
where $L_{rel}$ denotes the loss of the relative 3D hand pose estimation, $L_{view}$ denotes the loss of the view parameter estimation, $P_{i}^{rel}$ and $\hat{P_{i}^{rel}}$ denote the ground-truth and the estimation of the relative 3D hand pose, V and $\hat{V}$ denote the ground-truth and the estimation of the view parameter.

### 3.5. Hand Image Reconstruction

Hand image reconstruction [49] is employed as an auxiliary task to improve the generalization ability of the network. Following the idea of VAE, the hand image reconstruction module reconstructs the hand image using the sample *g* which is calculated using Equation (Equation 4). The same as [49], we apply a series of deconvolutional layers to reconstruct the hand image. The loss function of hand image reconstruction is defined as follows:(8)$$L_{recons}={\left\|I^{recons}-I^{rel}\right\|}_{1}+\frac{1}{2}\left(u^{T}u+sum\left(e^{\sigma }-\sigma -1\right)\right),$$
where the first term is the L1 distance between the original hand image and the reconstructed hand image, and the second term is the KL distance between the latent code probability distribution and the standard Gaussian distribution.

### 3.6. Semi-Supervised Learning

The total loss function of the training process is defined as follows:(9)$$L=\lambda_{1}L_{detection}+\lambda_{2}L_{gesture}+\lambda_{3}L_{rel}+\lambda_{4}L_{view}+\lambda_{5}L_{recons},$$
where the terms $L_{detection}$, $L_{gesture}$, $L_{rel}$, $L_{view}$ and $L_{recons}$ have been described before, and $\lambda_{i}\;\left(i=1\cdots 5\right)$ are the balancing weights whose values can be set to 1 or 0. The term $L_{detection}$ requires 2D hand location annotation, the term $L_{gesture}$ requires the gesture category annotation, the term $L_{rel}$ requires the relative hand pose estimation annotation, term $L_{view}$ requires the full hand pose annotation, and the term $L_{recons}$ requires no annotation.

Existing datasets contain either hand gesture recognition annotation or hand pose estimation annotation, and it is difficult to find a dataset which contains all the annotations mentioned above. To tackle the problem of lacking annotation, we adopt a semi-supervised learning scheme. For datasets with different annotations, the balancing weights $\lambda_{i}$ will be switched on or off accordingly. Specifically, for the images with detection annotation, the weight $\lambda_{1}$ is set to 1, otherwise 0. For the images with gesture category annotation, the weight $\lambda_{2}$ is set to 1, otherwise 0. As a specific hand gesture is associated with a specific relative hand pose, the weight $\lambda_{3}$ is set to 1 when the gesture annotation is available. For the images with hand pose annotation, the weights $\lambda_{3}$ and $\lambda_{4}$ are set to 1, otherwise 0. The weight $\lambda_{5}$ is always set to 1, because the hand image reconstruction requires no annotation.

## 4. CUG-Hand Dataset

The dataset contains 1757 color images, in which 1273 images are used for training, and 484 images are used for testing. The resolution of the image is 1280×720. The images are collected from 27 distinct subjects. The number of subjects on a single image varies from 1 to 7, which results in up to 8 hands per image. The dataset contains static ASL hand gestures, and the number of classes is 24 (the dynamic ASL gestures j and z are not included). In each image, there exist many background hands and a foreground hand performing an ASL hand gesture. In the training images, there are 5024 background hand instances and 1273 foreground hand instances. And in the testing images, there are 1485 background hand instances and 484 foreground hand instances. The area of the hand bounding boxes varies from 238 to 73,062 $pixel^{2}$. The CUG-Hand dataset provides the bounding boxes of all hand instances, and the gesture category of the foreground hands. The dataset does not have hand pose annotation.

## 5. Experiments

### 5.1. Experimental Setting

All the experiments are performed on a computer with a NVIDIA 1080Ti GPU. The proposed approach is implemented with PyTorch [55]. The network is trained using an Adam optimizer [56] with an initial learning rate of $1\times 10^{-3}$. The learning rate is multiplied by 0.1 every 10 epochs. The training process terminates after 20 epochs. The batch size is 32, and the input images are resized to a uniform resolution of 256×256. As the existing static hand gesture recognition datasets do not have hand pose annotation, we leverage the hand pose estimation knowledge by using the STB hand pose estimation dataset [57]. The STB dataset contains about 18 k stereo images with a resolution of 640 × 480, and the corresponding 3D hand pose annotations are provided.

### 5.2. Gesture Recognition on LaRED Dataset

The LaRED dataset [43] is a hand gesture recognition dataset. The dataset contains 27 basic gestures, and most of which are taken from American Sign Language. For each basic gesture, there are three different orientations, which results in totally 81 classes. Following the previous work [4], the metric AC is used to measure the hand gesture recognition accuracy. AC is the ratio of the number of samples correctly classified by the classifier to the total number of samples. The gesture recognition results of the compared methods are shown in Table 1. As the LaRED dataset is collected in constrained environment, the performance of the state-of-the-art on this dataset is approaching saturation. The AC of *Adam et al.* is 97.25%, and the AC of *Ours* is further improved to 99.96%. The accuracy of *Ours* is about 2.7 point higher than that of *Adam et al.*

### 5.3. Gesture Recognition on CUG-Hand Dataset

We collect the hand image patches (each image patch contains a single hand) in the CUG-Hand dataset, and recognize the gesture category of each image patch. In the experiments, following methods are compared: (1) *HOG+SVM* [58]; (2) *ResNet* [59]; (3) *Adam et al.* [4]; (4) *Baseline1*, gesture recognition based on the hand pose estimation result; (5) *Baseline2*, our approach without pose estimation and image reconstruction; (6) *Baseline3*, our approach without pose estimation; (7) *Baseline4*, our approach without image reconstruction; (8) *Ours*, our proposed approach.

The AC and the computational time per image of the compared methods are shown in Table 2. In the table, the column “Use Pose” denotes whether to use the hand pose estimation module. As the CUG-Hand dataset does not contain hand pose label, we learn the hand pose estimation knowledge from the STB hand pose estimation dataset using the semi-supervised learning scheme. And the column “Reconstruct” denotes whether to use the image reconstruction module.

*HOG+SVM*is a classical gesture recognition method, and its AC is 61.4%. *ResNet* is one of the most accurate image classification methods, and it outperforms the classical *HOG+SVM* by 24.3 points with the deep convolutional feature. *Adam et al.*is one of the most state-of-the-arts for hand gesture recognition, and it is accurate and efficient. The AC of *Adam et al.* is slightly lower than *ResNet*, but its computational efficiency per image is about 20 higher than that of *ResNet*. Overall, the experimental results show that, the AC of *Ours* is significantly higher than that of the compared methods by a large margin, and the computational time per image of *Ours* is also very efficient (5.7 ms per image, that is, about 175 frames per second).

*Baseline1*denotes conducting the gesture recognition directly based on the hand pose estimation result, and its AC is 64.8. As inaccurate hand pose estimation result leads to false gesture classification, the AC of *Baseline1* is much lower than that of other baselines. It is better to jointly learn the relationship between the hand pose and the hand gesture using intermediate level shared feature. The AC of *Baseline2* is 86.6. Comparing to *Baseline2*, the AC of *Baseline3* is improved by 1.2 points with the image reconstruction module, and the AC of *Baseline4* is improved by 2.4 pints with the hand pose estimation module. Comparing to *Baseline3*, the AC of *Ours* is improved by 3.3 points with the hand pose estimation module. The experimental results show that the hand pose estimation module significantly benefits the gesture recognition task.

The ROC curves of the compared methods are shown in Figure 3. The true positive rate is correlated with the false positive rate. The target is to achieve high true positive rate with low false positive rate. In other words, the closer the ROC curve be to the top-left corner, the better the ROC performance is. As can be seen in the figure, the ROC performance of Ours is better than that of the compared methods. The gesture recognition confusion matrix of *Ours* is shown in Figure 4. The x-axis denotes the predicted gesture category of the sample, and the y-axis denotes the ground-truth of the sample. The higher probability on the diagonal of the confusion matrix is, the more accurate the gesture recognition is. As it can be seen, most of the gesture categories can be accurately classified, while some gestures are more difficult to be recognized than the others. In the ASL alphabet, some static hand gestures are very similar to other gestures, for example, “K”, “V”, “M”, “N”, “S”, “T”. It is hard to distinguish these “difficult” gestures when the distance from the hand to the camera is not near or the lightening condition is not well. For example, the gesture “M” may be falsely classified as “N” by a rate of 0.15.

By leveraging the hand pose knowledge in the STB dataset, the proposed method can predict the gesture as well as the 3D joints/skeletons of hand images in CUG-Hand dataset which does not have hand pose annotation. The 3D hand pose estimation results of *Ours* are visually shown in Figure 5. The 3D joints are projected onto the cropped image plane, and the more tightly the skeletons align to the hand, the more accurate the estimated pose is. As can be seen in Figure 5a, the predicted 3D skeletons can well align to the the hands in image. And the failure cases are shown in Figure 5b.

### 5.4. Gesture Detection on CUG-Hand Dataset

In complex unconstrained environments, there may exist multiple hands. We detect all hand instances with different gesture categories in unconstrained environments. The gesture detection accuracy is evaluated using the mAP metric defined in the object detection field [51]. Firstly, we calculate the Average Precision (AP) of each class with an IOU threshold of 0.5, and then the mean AP of all classes is defined as
(10)$$\mathrm{mAP}=\frac{1}{N}\sum_{i=1}^{N}{\mathrm{AP}}_{i},$$
where *N* denotes the number of classes, *i* denotes the class index, and ${\mathrm{AP}}_{i}$ denotes the AP of class *i*. In this study, N=25, that is, 24 ASL hand gestures and background hand. The following methods are compared: (1) *FasterRCNN* [51], one of the most widely used object detection baselines in the computer vision community; (2) *Adam et al.* [4]; (3) *Ours*, the proposed approach.

The mAP of the compared methods are shown in Table 3. The hand gesture detection accuracy of *Ours* is the highest among that of the compared methods. The mAP of *FasterRCNN* is 63.5, and that of *Ours* is 18.8 points higher than that of *FasterRCNN*. The mAP of *Ours* is also 12.1 points higher than that of *Adam et.al.* The precision, recall, and F1 score of *Ours* are 87.5, 75, and 80.7 respectively, which are significantly higher than that of *FasterRCNN* and *Adam et al.* The Precision Recall (PR) curves of the compared methods are shown in Figure 6. The detection AP of *Ours* for each gesture category is shown in Table 4. And the hand gesture detection and pose estimation results of *Ours* are visually presented in Figure 7. By leveraging the hand pose estimation knowledge, the detection accuracy of the proposed approach is significantly improved.

The hand detection task is correlated to the human body parts detection task. When both hands and body parts annotations are available, exploring the relationship between hands and body parts will be helpful to improve the hand detection precision. However, existing hand gesture datasets normally contain hand annotation only, therefore our approach detects hands without explicitly exploring the information of other body parts. As the hand detection does not rely on the body part detection, it can successfully work when only hand appears (i.e., no other body part appears). Our hand detector inherits from our previous works [49,60]. In [60] it is shown that, our hand detection scheme works better than OpenPose [61] (a famous human body estimator which detects all body parts) in terms of hand detection, because OpenPose cannot correctly detect hands when other body parts do not appear. The relationship between hand detection and body parts detection is an interesting topic, and we will consider to study this topic in our future work.

## 6. Conclusions

We propose a hand gesture recognition approach by joint learning a shared feature for hand gesture recognition and hand pose estimation tasks. To overcome the problem of lacking annotation, the semi-supervised training scheme is used to benefit the hand gesture recognition task from hand images without hand gesture annotation. The experimental results show that, the proposed method effectively leverages the hand pose estimation knowledge for hand gesture recognition, and the hand image reconstruction task further improves performance. Comparing to *Baseline1* which recognizes the gesture directly based on the pose estimation result, the proposed approach significantly improves the accuracy by a large margin. Comparing to *Baseline2*, the hand pose estimation and the hand image reconstruction tasks together improve the accuracy by 5.2%. Comparing to *Baseline3*, the hand pose estimation task improves the accuracy by 3.8%. Comparing to *Baseline4*, the hand image reconstruction task improves the accuracy by 2.4%. Furthermore, the proposed approach can detect foreground hand, recognize the hand gesture, and estimate the hand pose simultaneously in unconstrained environments. In the future, we plan to study the dynamic hand gesture recognition, and also the interaction between hand and object.

## Figures and Tables

**Figure 1 sensors-21-01007-f001:**
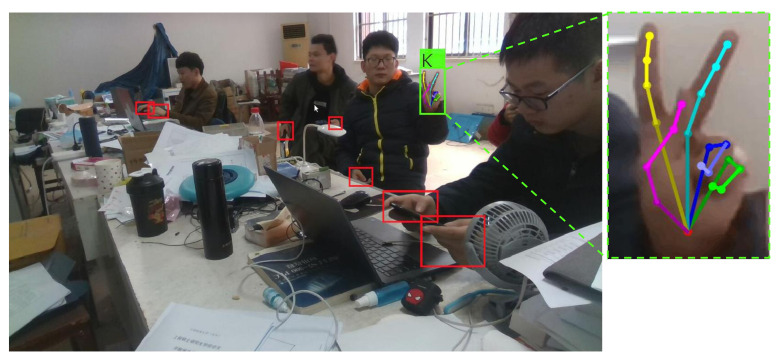
The proposed approach detects hands, recognizes the foreground hand gesture, and estimates the hand pose simultaneously. The red boxes denote the detected background hands, the green box denotes the detected foreground hand which is performing a gesture, the label attached to green box denotes the recognized gesture, and the right figure zoom in on the corresponding hand pose estimation result.

**Figure 2 sensors-21-01007-f002:**
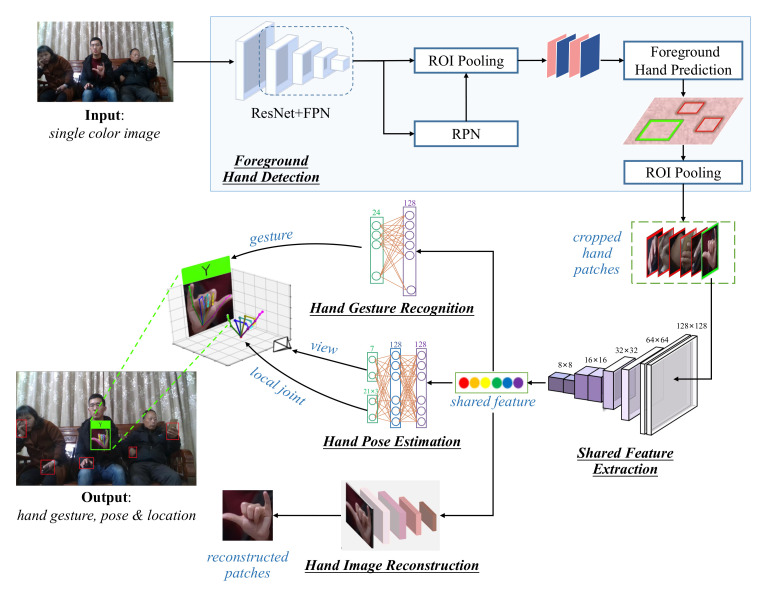
The framework of the proposed approach.

**Figure 3 sensors-21-01007-f003:**
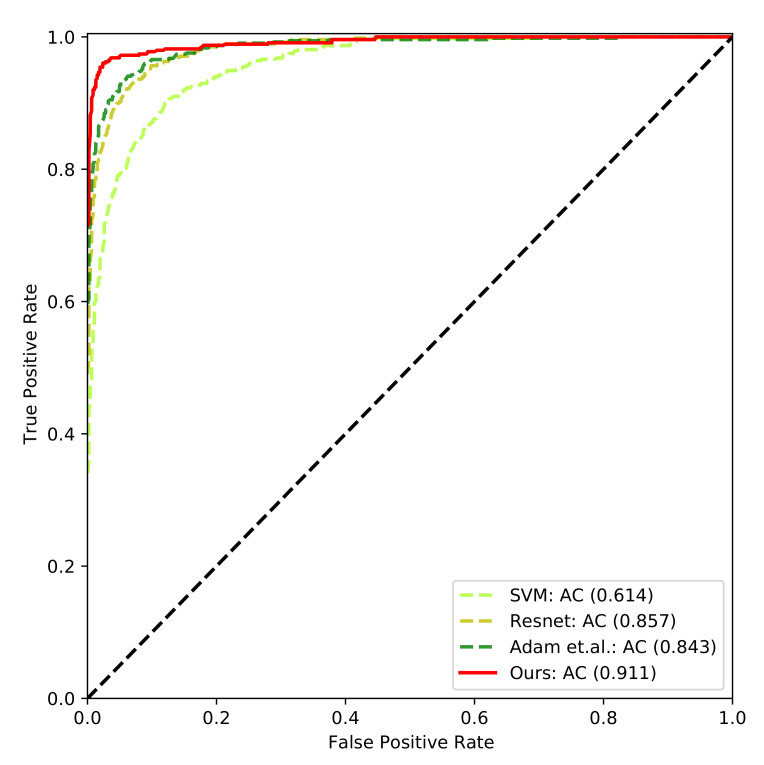
The ROC curve of the compared methods.

**Figure 4 sensors-21-01007-f004:**
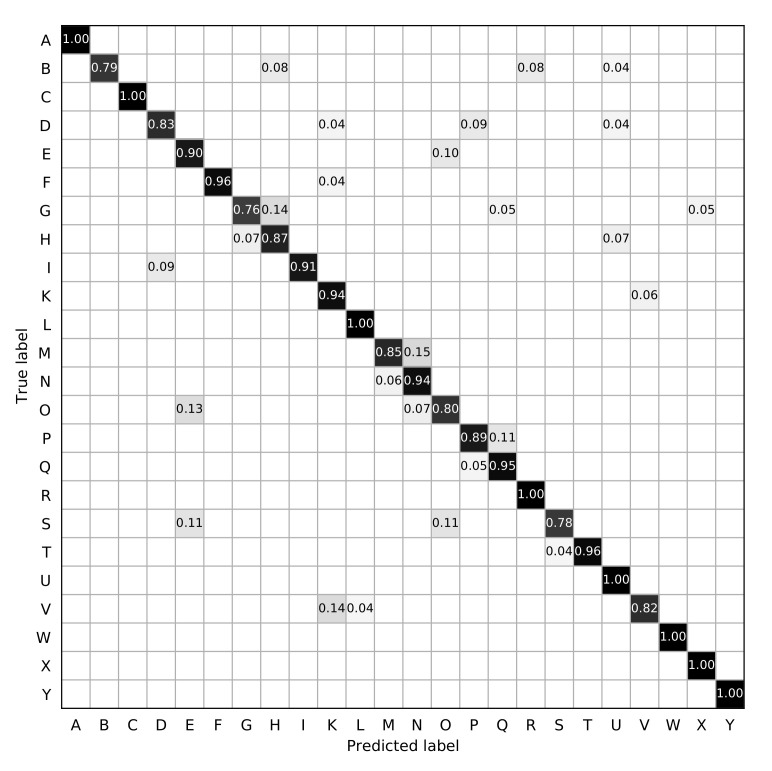
The confusion matrix of *Ours*.

**Figure 5 sensors-21-01007-f005:**
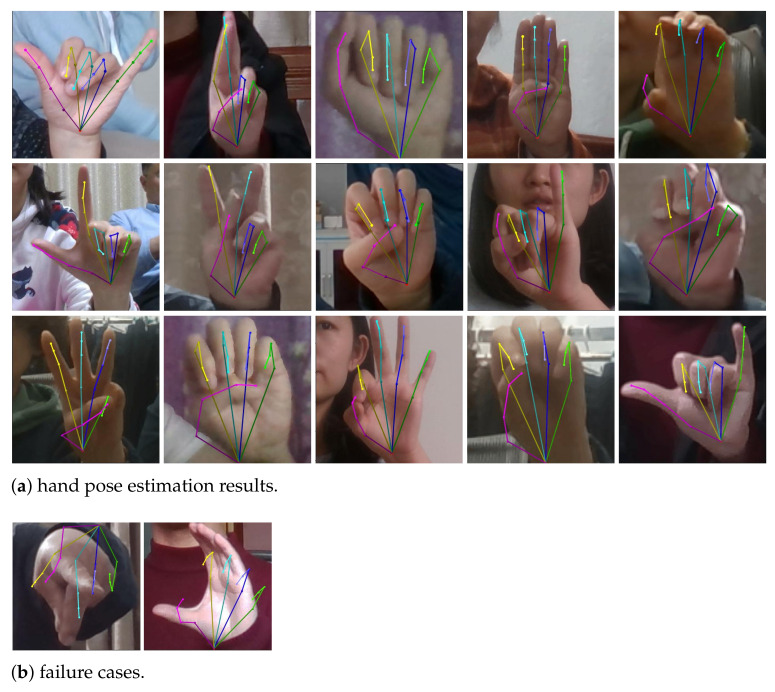
Hand pose estimation results on CUG-Hand dataset.

**Figure 6 sensors-21-01007-f006:**
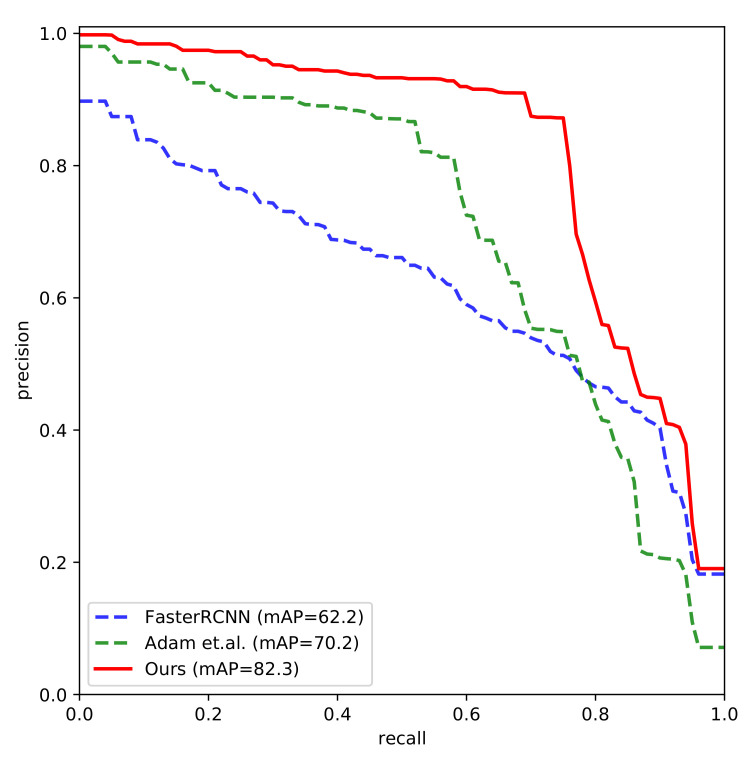
The Precision Recall (PR) curves of the compared methods.

**Figure 7 sensors-21-01007-f007:**
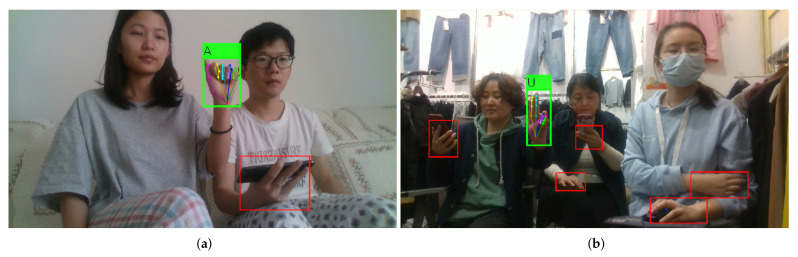
Hand gesture detection and pose estimation results of *Ours*. The green boxes denote the detected foreground hands, the green labels attached to the green boxes denote the recognized gesture, and the red boxes denote the detected background hands. The 3D skeletons of foreground hands are projected on the 2D image plane.

**Table 1 sensors-21-01007-t001:** Gesture recognition results on LaRED dataset.

Methods on LaRED Datasets	AC (%)
*SVM*	73.86
*DBN*	74.90
*SAE*	86.57
*Adam et al.*	97.25
*Ours*	99.96

**Table 2 sensors-21-01007-t002:** Gesture recognition accuracy and efficiency of the compared methods. The column “Use Pose” denotes whether to use the hand pose estimation module. The column “Reconstruct” denotes whether to reconstruct the image reconstruction module.

Methods	Use Pose	Reconstruct	AC (%)	Time Per Image (ms)
*HOG+SVM*	×	×	61.4	11.7
*ResNet*	×	×	85.7	51.7
*Adam et al.*	×	×	84.3	2.5
*Baseline1*	*√*	×	64.8	6.0
*Baseline2*	×	×	86.6	4.7
*Baseline3*	×	*√*	87.8	5.3
*Baseline4*	*√*	×	89.0	4.8
*Ours*	*√*	*√*	91.1	5.7

**Table 3 sensors-21-01007-t003:** The hand gesture detection mAP of the compared methods. The bold font means the best score.

	mAP (%)	Precision (%)	Recall (%)	F1 Score (%)
*FasterRCNN*	63.5	53.3	72	61.3
*Adam et al.*	70.2	81.3	58	67.7
*Ours*	**82.3**	**87.2**	**75**	**80.7**

**Table 4 sensors-21-01007-t004:** Detection Average Precision (AP) of *Ours* for each hand gesture category. “*∅*” denotes the background hand.

Detection AP of *Ours*				
A	B	C	D	E
93.2	94.1	94.1	76.2	69.9
F	G	H	I	K
100.0	72.1	66.7	92.6	80.8
L	M	N	O	P
75.4	64.7	69.8	69	66.5
Q	R	S	T	U
72.4	86.2	91.2	74.1	84.9
V	W	X	Y	*∅*
89.6	94.1	97.6	100.0	90.0

## Data Availability

The data that support the findings of this study will be released on https://github.com/waterai12/CUG-Hand-Gesture.

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
