# Peer review of "Semi-Supervised Joint Learning for Hand Gesture Recognition from a Single Color Image"

_sensors, 2021, doi:10.3390/s21031007_

Round 1

Reviewer 1 Report

The paper addresses the difficult problem of hand pose estimation and hand gesture recognition in cluttered environments. The problem is reduced to a few static gestures (a part of the Sign Language Alphabet) for single color images.
The paper proposes an interesting method composed of six main parts : foreground /background hand detection with FPN and RPN, lightweight CNN network for shared feature extraction, hand gesture recognition, hand pose estimation, hand image reconstruction, semi-supervised learning, where the five previous objective functions are fused
A new dataset is proposed, in which the images are collected in cluttered environments, and the number of hands per image is up to 8. The code and dataset related to the paper will be made available online.
In experiments, the method is compared to other techniques and baselines. The proposed method obtains convincing results in terms of accuracy and efficiency (execution times).
The paper is well written.

The term static hand gesture could be replaced by handshape, which is the term used in sign languages.

Reviewer 2 Report

The authors of the article describe a new hand gesture recognition approach by joint learning a shared feature, for hand gesture recognition and hand pose estimation. The idea of article seems to be an innovative one, the experimental results showing that the proposed method effectively leverages the hand pose estimation knowledge for hand gesture recognition, and the hand image reconstruction task further improves performances of the recognition.

The article is well written, it is splitted into appropriate parts and easy to track and understand. The scientific contributions are also significant in qualitative and quantitative aspects.

One minor comment:

Regarding the Results sections, the experimental results should be grouped together under the same type of hand gesture recognition results for all image databases used. In this way, the reader can more easily perform a general comparative analysis of the results obtained in the experimental section.

Reviewer 3 Report

The work in question deals with the recognition of static gestures on the basis of a single color image, using deep learning. The gestures are the 24 characters of ASL sign language denoting letters. As recognition applies to static gestures, gestures denoting J and Z that require movement are omitted. Gesture recognition is combined with hand pose recognition, which facilitates the entire recognition process. Input images do not need to be specially prepared - the system works with any background and hand position in the photo. It is also possible to analyze images containing more than one hand. Cases of up to eight hands in one photo have been tested. One of such detected hands is considered as “foreground” the gesture presented by it is recognized. The recognition process consists of several stages: finding (detecting) the hand in the analyzed image, determining the features, recognizing the gesture, determining the hand pose, and reconstructing the image of the hand. Reconstruction allows, among others to verify the correctness of the recognition of a gesture by putting on the image the projection of the reconstructed skeleton of the hand presenting the recognized gesture. The authors described in detail the proposed system and the method of creating a database of images for training and tests. The problem was finding an annotated set if images containing annotations for both the gestures and hand poses.

Very good results of the classification correctness were obtained. According to the data from Table 1, they are superior to all the methods with which the authors compared themselves.

It seems interesting to combine hand detection with full-body detection and position determination. This would help you find your hand correctly as in figure 7f. Perhaps the method used implicitly finds the configuration of the body. It would be interesting to mention this and possibly to check how hand detection works after removing the image of the rest of the person's body. This would determine whether hand detection is performed solely on the basis of an image of that hand or requires the presence of an image of a larger part of the body. The article mentions that some hand images were only 238 pixels in size, resulting in a 14x17 pixel bounding box. The presented results are very interesting. I propose to extend the method for recognizing dynamic gestures, which is necessary (apart from enlarging the dictionary of recognized gestures) when recognizing sign language.

Reviewer 4 Report

Dear Authors:

"Semi-supervised joint learning for Hand Gesture Recognition from a Single Color Image" is a good written manuscript.

Overall, it was easy to follow the article. My specific comments are given below:

  • Line 250, add a space between "lambda3 and is". This is also applicable if there are other similar instances
  • For Confusion marix, would you provide a discussion on why certain letters were predicted with a high accuracy and others don't.
  • For conclusion, while comparing your accuracy with other baseline methods, you mentioned that your method is X point more than a baseline. Instead, I would recommend to report improvement as % from the baseline. For example, mention that comparing to Baseline4, the hand image reconstruction task improves the accuracy by 2.4% (91.1-89)/89 percentage. Please rewrite all other improvement as % of improvement from Baseline.
  • All the best!
